# Investigation of Curing Process and Thermal Behavior of Copolymers Based on Polypropylene Glycol Fumarate and Acrylic Acid Using the Methods of DSC and TGA

**DOI:** 10.3390/polym15183753

**Published:** 2023-09-14

**Authors:** Gulsym Burkeyeva, Anna Kovaleva, Yerkeblan Tazhbayev, Zhansaya Ibrayeva, Lyazzat Zhaparova

**Affiliations:** The Department of Organic Chemistry and Polymers, Chemistry Faculty, Karaganda University of the Name of Academician E.A. Buketov, Karaganda 100024, Kazakhstan; guls_b@mail.ru (G.B.); cherry-girl1899@mail.ru (A.K.); tazhbaev@mail.ru (Y.T.); lyazzh@mail.ru (L.Z.)

**Keywords:** unsaturated polyesters, acrylic acid, curing, DSC, kinetics, TGA, thermostability, activation energy

## Abstract

In this work, the possibility of preparation of copolymers of three-dimensional crosslinked structure based on polypropylene glycol fumarate and acrylic acid is shown. The initial reagent polypropylene glycol fumarate has been synthesized by polycondensation reaction of fumaric acid and propylene glycol. The curing process of polypropylene glycol fumarate and acrylic acid at various mole concentrations was studied using DSC method at isothermal and dynamic regimens. Curing in isothermal condition was carried out at temperatures of 60 °C, 70 °C, and 80 °C. Residual reactivity was evaluated at a dynamic regimen within the temperature range from 30 °C to 200 °C at a constant heating rate. On the basis of calorimetric studies, the thermal effects and kinetic parameters of the reaction (conversion, reaction rate, activation energy) have been determined. Thermal behavior of cured samples of p-PGF-AA was estimated using dynamic thermogravimetry (TGA). According to TGA data, the process of decomposition of the studied copolymers proceeds in several stages. Based on the results obtained, the activation energies of thermal decomposition were calculated using the iso-conversional methods of Kissinger–Akakhira–Sunose and Friedman.

## 1. Introduction

Unsaturated polyester resins are one type of primary thermosetting materials. The unsaturated polyester resins are peculiar due to their availability and good processability into various polymeric, compositional materials and reinforced plastics [1]. In particular, unsaturated polyesters are widely used as binding agents when obtaining compositional materials, including the goods of structural supplies for the preparation of forming masses, laminates, coatings, and glues.

Unsaturated polyesters are of special interest among the compounds, which are “convenient” for the modification. In industry, unsaturated polyesters are obtained by polycondensation of unsaturated dicarboxylic acids with polyatomic alcohols [2]. Owing to the presence of a reactive double bond, the unsaturated polyesters are able to enter the copolymerization reaction with many monomers forming cured reaction products. Obtained cured unsaturated polyester resins are the materials which possess high durability, wear-resistance, very good dielectric properties, high chemical stability to the various mediums, and adaptability to manufacture at a wide range of temperatures [3,4].

When creating the polymeric binders with given properties, great attention is paid to the investigation of the processes of their curing since the peculiarities of the process of structuring define the technology of preparation of binding agents, degree of cure (i.e., the inalterability of the properties during operational process), and final operational characteristics. The reprocessing of unsaturated polyester resins requires the understanding of the kinetics of the copolymerization reaction during the process of curing. The kinetics of curing of unsaturated polyester resins is of great value for optimization of many industrial processes with the participation of unsaturated polyesters. In addition, when working out of the new compositions of unsaturated polyesters and control of the quality of manufactured product, there is a necessity to study their thermal properties.

Most of parts of the research work in the field of unsaturated polyesters are devoted to the synthesis and study of their properties cured with styrene [5,6,7]. At the same time, there are no data on studying the curing process of unsaturated polyesters with unsaturated carboxylic acids using the method of DSC and their thermostability. There are data on the investigation of thermal stability of the copolymers of unsaturated polyester resins with hydrophobic monomers at a wide range of temperatures in the literature [8]. The results of thermogravimetric analysis have shown that these copolymers are thermostable until 250–300 °C, but after that, the thermal destruction which goes on several stages takes place [9].

In this regard, the aim of this work is the investigation of the curing process of polypropylene glycol fumarate with acrylic acid using the DSC method and the study of thermal properties of synthesized copolymers with the help of TGA.

## 2. Materials and Methods

### 2.1. Materials

The following reagents were used in this work: fumaric acid and acrylic acid (AA) (“Vekton”, Saint-Petersburg, Russia); propylene glycol (99.8%) (“Ekos-1”, Moscow, Russia); catalyst zinc chloride (“Reachem”, Moscow, Russia); benzoyl peroxide (BP) Luperox A75 (“Sigma-Aldrich”, Burlington, MA, USA); and accelerator dimethyl aniline (DMA) 99.5% (“Chemical line”, Saint-Petersburg, Russia).

### 2.2. Preparation of Polypropylene Glycol Fumarate

Initial polypropylene glycol fumarate (*p*-PGF) is an unsaturated polyester, which has been synthesized by polycondensation of fumaric acid with propylene glycol at a temperature of 423–433 K [10,11]. The reaction of synthesis and the most probable structure of the obtained polyester is presented in Figure 1.

### 2.3. Curing of Polypropylene Glycol Fumarate

Curing of the reaction mixture of *p*-PGF with AA was carried out by radical copolymerization in mass at various mole ratios of the comonomers at temperatures of 60 °C, 70 °C, and 80 °C in the presence of the initiator BP. The concentration of the initiator BP was 1% on a mass of initial mixture of *p*-PGF-AA for all the experiments.

### 2.4. Methods

#### 2.4.1. Gel-Permeation Chromatography

Molecular mass of *p*-PGF was determined using gel-permeation chromatography (GPC) on a chromatograph Malvern equipped with double detector Viscotek 270 (Malvern Instruments INC, Bristol, United Kingdom). According to the results of the experiment, the molecular weight of the polymer was Mw—2630 Da. Polystyrene was used as a standard, and dioxane was used as a solvent. Data were found out with the help of a light scattering (LS) detector (Malvern Instruments INC, Bristol, United Kingdom). The experiment was carried out by universal calibration (UC), the dn/dc index of the sample was 0.0839, the column type was T6000M for organic solvents, the current rate was 1.0 mL/min, and the temperature was 40 °C.

#### 2.4.2. Differential Scanning Calorimetry (DSC)

Measurements of the heat effects of the curing reaction of *p*-PGF-AA and of residual reactivity were carried out using the DSC method on the device of synchronic thermal analysis, Labsys Evolution TG-DTA/DSC (Setaram Instrumentation, Caluire, France). The reaction mixture with the mass of 95–100 mg consisting of *p*-PGF with AA and initiator BP (the content is 1% on the mass of initial mixture) in opened corundum crucibles was placed into the measurement cell of the calorimeter. Nitrogen was used as a purging gas. The tests were performed in isothermal regimen at temperatures of 60 °C, 70 °C, and 80 °C.

After finishing the isothermal experiment, the DSC cell was quickly cooled to room temperature at the rate of 100 °C/min under running water (15–17 °C). After stabilization of the temperature, the residual heat of the curing reaction was measured until the exothermal reaction was no longer observed. The residual heat of the reaction was registered in a dynamic regimen at the heating rate of 10 °C/min within the temperature range of 30 °C to 200 °C.

When studying the kinetics of isothermal curing of thermosetting resins using DSC, it is supposed that the amount of heat liberated is proportional to the degree of cure (α) of the sample and the reaction rate is proportional to the heat flow measured [12,13]. The heat flow (as a function of the dependence on time) is registered with the help of a device. Thus, the conversion was determined according to the equation given below (α):(1)$$\alpha =\frac{\Delta H_{t}}{\Delta H_{tot}},$$
where $\Delta H_{t}$—the heat of the reaction at a time moment t, (J/g); $\Delta H_{tot}$—total heat of the reaction, (J/g).

Total heat of curing reaction $\Delta H_{tot}$:(2)$$\Delta H_{tot}=\Delta H_{iso}+\Delta H_{r}$$
where $\Delta H_{iso}$—heat liberated at isothermal regimen of curing; $\Delta H_{r}$—is the residual heat released when the sample was post-cured in a dynamic DSC test in the temperature range of 30–200 °C after the first isothermal cure at a reaction rate of 10 °C/min.

Consequently, the reaction rate is expressed by the equation:(3)$$\frac{d\alpha }{dt}=\frac{dH\left(t\right)}{dH_{tot}dt}=k\left(T\right)f\left(\alpha \right),$$
where $dH\left(t\right)/dt$—the rate of the heat flow; $k\left(T\right)$—the reaction rate constant.

The reaction rate constant $k\left(T\right)$ is expressed by the Arrhenius equation:(4)$$k\left(T\right)=Ae^{\left(\frac{-E_{a}}{RT}\right)},$$

Converting Equation (3):(5)$$\ln k=\ln A-\frac{E_{a}}{RT}$$
(6)$$E_{a}=-\tan\alpha R$$
where

$E_{a}$—activation energy of the reaction, kJ/mol;

A—pre-exponential factor;

R—universal gas constant (J/mol·K);

T—temperature, K.

The activation energy, $E_{a}$, is defined according to a tangent of angle of slope of the dependence $\ln k=f\left(1/T\right)\;$ using Equations (5) and (6).

#### 2.4.3. Thermogravimetric Analysis (TGA)

The investigation of the thermal properties of the copolymers of *p*-PGF-AA was carried out on a device for synchronic thermal analysis, Labsys Evolution TG-DTA/DSC (Setaram Instrumentation, Caluire, France), in a dynamic regimen at the temperature interval of 30–600 °C. The samples were heated in the crucibles made of Al_2_O_3_ at the rates of 2.5, 5, 10, and 20 °C/min in the nitrogen medium at the heating rate of 30 mL/min. The calibration of the device for thermogravimetric studies and heat flow was conducted using the CaCO_3_ and In (Indiy) standards, correspondingly.

The kinetics of thermal decomposition are usually expressed by the combination of the Equations (2) and (3):(7)$$\frac{d\alpha }{dt}=Aexp^{-\frac{Ea}{RT}}\cdot f\left(\alpha \right),$$
the obtained equation gives the foundation for the differential kinetic methods. Since it is not possible to solve the right part of the Equation (7) analytically, different approximate methods are used in practice.

The differential method of Friedman was obtained using the iso-conversion approach to Equation (7), which results in [14]:(8)$$\ln\left(\frac{d\alpha }{dt}\right){\;}_{\alpha ,\;i}=\ln\left[f\left(\alpha \right)A_{\alpha }\right]-\frac{E_{a}}{RT_{\alpha ,i}},$$

The index, *i*, is introduced for designation of various temperature programs. *T_α,i_*—is a temperature when the conversion (α) is reached at temperature program, *i*. As a result, Equation (8) supposes that *T_α,i_* changes linearly with time passing with the heating rate β_i_.

There are number of integral iso-conversional methods which differ on their approximation of temperature integral in Equation (8). The approximation of Murrey and White leads to B = 2 and C = 1, and it results in another known equation, which is often called the equation of Kissinger–Akakhira–Sunose [15,16]:(9)$$\ln\left(\frac{\beta_{i}}{T_{\alpha ,i}^{2}}\right)=\mathrm{Const}-\frac{E_{a}}{RT_{\alpha ,i}},$$
activation energy for the different conversion degrees is calculated from the slope of the dependence $\ln\left(\frac{\beta_{i}}{T_{\alpha ,i}^{2}}\right)$ on 1/T.

#### 2.4.4. HPLC Method, IR-Spectroscopy, Scanning Electron Microscope (SEM)

The composition of cured samples of *p*-PGF-AA was defined using the HPLC (high performance liquid chromatography) method on an HPLC20 (Shimadzu, Kioto, Japan) by analyzing the mother liquors according to the residual principle.

Identification of cured products of *p*-PGF-AA was performed using IR-spectroscopy. The IR-spectra of the samples were registered on KBr tablets on spectrometer FSM 1201 (Infraspek, Saint-Petersburg, Russia) [17,18].

Electron-microscopic studies of cured samples of *p*-PGF-AA have been carried out on scanning electron microscope (SEM) MIRA 3 (Tescan, Brno, Czech Republic) at an accelerating voltage of 5 kV.

## 3. Results and Discussion

### 3.1. Curing of p-PGF-AA

The curing reaction of *p*-PGF-AA is free radical polymerization of active chains formed by the bond scission of double bonds of *p*-PGF and the molecules of AA (Figure 2). During the copolymerization (curing) of *p*-PGF with AA, there are four elementary reactions that take place: homopolymerization of AA, adjunction of the molecules of AA to the active centers of *p*-PGF, combination of two molecules of unsaturated polyester according to their active centers, and crosslinking of polyester centers of AA by chain. As a result of all these reactions, a three-dimensional network is formed. The scheme of radical copolymerization is presented in Figure 2.

#### Isothermal and Dynamic DSC Results

With the aim of investigating the kinetic parameters of the curing process of *p*-PGF-AA, the DSC studies have been carried out.

In Figure 3 and Figure 4, there are DSC curves of the cure of *p*-PGF-AA of the compositions of 30.05:69.95 mol.%, 45.13:54.87 mol.%, and 61.21:39.79 mol.% given.

The thermograms of curing of the studied samples in isothermal regimen at temperatures of 60 °C, 70 °C, and 80 °C are shown in Figure 3. From the results obtained, it follows that the composition and the temperature have influence on the reaction rate of curing of *p*-PGF-AA. In Table 1, the data on the induction time t_i_, curing time t_curing_ at a maximum meaning of dα/dt and the time of vitrification t_v_ are given. The vitrification time was determined as the time when the curing reaction is finished at isothermal conditions, i.e., the isothermal thermogram then goes back to its baseline.

The data in Table 1 point to a decrease of the curing time with the increasing of temperature and the higher the temperature, the higher the reaction rate. In addition, the reaction rate is affected by the concentration of the initial reagents of *p*-PGF and AA. An explanation for such dependence is the comparison of the activity constants (r_1_,r_2_) of the initial co-reagents of *p*-PGF and AA. The activity constants of the initial co-reagents calculated using the integral method of Maiyo-Luis, is r_1_ = 0.82 for *p*-PGF and r_2_ = 1.21 for AA [19]. As a consequence, it can be concluded that *p*-PGF is less reactive comparing to AA, and correspondingly, the curing time and the rate of copolymerization reaction drop with the rise of the concentration of *p*-PGF. In addition, the following calorimetric studies of isothermally cured copolymers with high content of unsaturated polyester in a dynamic regimen have shown the liberation of the residual heat. Thus, determination of the residual heat was carried out after isothermal cure of *p*-PGF-AA by cooling the system followed by its heating at dynamic regimens (the “heat–cool–heat” procedure). Figure 4 presents DSC thermograms that illustrate the residual exothermic effect.

Thus, Table 2 has the values of heat of the reaction in isothermal and dynamic regimens of curing. From the experimental data of DSC, it is clear that with the temperature growth, the isothermal heat of curing rises ∆H_iso_ (Figure 3), and the residual heat, ∆H_r_, which is registered in a dynamic regimen (Figure 4), decreases. It is known from the literature [20], that the sum of ∆H_iso_ and ∆H_r_ gives the total amount of heat, ∆H_tot_, of the reaction for the given system.

From the DSC curves (Figure 4), it is obvious that a considerable residual heat was observed in the samples with high content of *p*-PGF (~45–60 mol.%), which was isothermally cured at low temperatures. The copolymers with the ~45–60 mol.% amount of *p*-PGF, which were isothermally cured at the temperatures of 60 °C, 70 °C, and 80 °C in a dynamic regimen, have shown the liberation of residual heat. Thus, the sample *p*-PGF-AA of the composition 61.21:39.79 mol.% is characterized by maximum values of residual enthalpy curing ($\Delta H_{r}$). It points out that for the copolymers with the higher content of *p*-PGF, the isothermal cures at 60 °C, 70 °C, and 80 °C is not carried out completely, whereas the sample with the minimum amount of *p*-PGF 30.05:69.95 mol.% at an isothermal regimen is cured completely. It is confirmed by the minimum meanings of the residual heat of curing, which are obtained at a dynamic regimen of DSC. Thus, the sample with the composition of 30.05:69.95 mol.%, which is isothermally cured at 60 °C, is characterized by the liberation of minimum quantity of residual heat of 1.687 J/g, and at the temperatures of 70 °C and 80 °C, the heat release is almost not observed (0.827 and 0.078 J/g, accordingly). Incomplete cure of the copolymers of *p*-PGF-AA with the high content of unsaturated polyester at low temperatures takes place due to vitrification, which complicates the interaction of reactional groups and prevents a full cure [21].

Table 3 has the results of the dependence of conversion and constant of rate and activation energy on the composition of reaction mixture and temperature presented. From the experimental data, it is seen that with the increasing temperature of isothermal curing, the conversion calculated according to the Equations (1) and (2), rises. Thus, *p*-PGF-AA 45.13:54.87 mol.% at a temperature of 60 °C had a ~85% conversion, while the degree of conversion goes up and reaches the values up to ~97% and 99% at 70 °C and 80 °C, correspondingly. Thus, the sample of the composition of *p*-PGF-AA 45.13:54.87 mol.% at a temperature of 60 °C had a ~85% conversion, while at 70 °C and 80 °C the degree of conversion goes up and reaches values up to ~97% and 99%, correspondingly. The isothermal cure of the sample of *p*-PGF-AA with the content of 30.05:69.95 mol.% within the whole range of the temperatures is characterized by the maximum values of conversion ~99–100%.

The obtained cured samples have been identified with the help of IR-spectroscopy. 

As it is seen from Table 3, with the increase of the amount of *p*-PGF in the reaction mixture, an apparent activation energy, E_a_, grows, reaching the meaning of 77.3 kJ/mol. At the same time, the rate of cure of the systems studied falls. Thus, the cure rate of *p*-PGF-AA of the composition of 61.21:39.79 mol.% is lower, and accordingly, the curing process goes slower. The constants of the rate of reaction obtained with the help of Equation (3) are shown in Table 3.

The activation energy of the system based on *p*-PGF-AA of the composition of 30.05:69.95 mol.% is less than for *p*-PGF-AA with the composition of 45.13:54.87 mol.% and 61.21:39.79 mol.% and is equal to 27.4 kJ/mol at the same curing conditions. The difference in the values of E_a_ of the systems investigated is explained by the growth of the quantity of the vinyl monomer of AA in the reaction mixture and its higher activity in the reactions of radical copolymerization in comparison to *p*-PGF.

Figure 5 presents the results of IR-spectra of the initial *p*-PGF and its copolymers with AA of the composition of 45.13:54.87 mol.%: *p*-PGF-AA* (the sample obtained as a result of isothermal curing at 60 °C) and *p*-PGF-AA (the sample of post-curing with the conversion degree of ~85%).

The analysis of IR spectra of the initial *p*-PGF points to the presence of characteristic bars appearing within the interval of 1570–1590 cm^−1^, which corresponds to unsaturated double bonds of polyester. There are also absorption bars within the range of 2860–2885 cm^−1^, which correspond to methyl group—CH_3_ in *p*-PGF. The appearance of intensive narrow bars in the range of 1400–1440 cm^−1^ is characteristic for –CH_2_–CO–. The presence of ester-group –COOC=C– is confirmed by the peaks at 1778 cm^−1^ and 1792 cm^−1^.

As a result of the curing reaction of the initial *p*-PGF of acrylic acid accompanied by the breakdown of the double bond and the formation of three-dimensional polymeric network, the IR-spectra of *p*-PGF-AA* and *p*-PGF-AA has the peaks at 1576 cm^−1^ and 1579 cm^−1^, which correspond to some quantity of unreacted unsaturated double bonds of polyester. In this case, the area of these peaks shrinks during the process of curing, which is explained by the growth of the numbers of the addition reactions of the chains of AA to *p*-PGF on double bond during the reaction of curing. The appearance of the peak at 1719 cm^−1^ in the IR-spectra of the copolymer of *p*-PGF-AA* and the peaks at 1702 cm^−1^ and 1724 cm^−1^ in the spectra of the copolymer of *p*-PGF-AA confirm the presence of carboxylic groups –COOH, whereas the presence of the peaks at 2859 cm^−1^ and 2915 cm^−1^ characterize the methylene groups –CH_2_– of acrylic acid. It is noteworthy that the intensity of these peaks increases within the process of complete curing due to the increase of the amount of the chains of AA in the composition of the final reaction products.

The morphology of the surfaces of cured samples of *p*-PGF-AA was studied with the help of scanning electron microscopy (Figure 6). Images were obtained using a secondary electron detector (SE detector) at an accelerating voltage (HV) of 5 kV.

There are some differences of morphological surfaces of the samples on the microphotographs. From microanalysis, it is seen that the copolymers with the high content of *p*-PGF (~45–60 mol.%) is characterized by the dense structure with the smaller porous diameter in comparison with the sample with the composition of 30.05:69.95 mol.%.

### 3.2. Thermogravimetric Analysis (TGA)

Further investigations of thermal properties of synthesized copolymers of *p*-PGF-AA have been conducted. In this regard a dynamic thermogravimetric analysis of cured samples of *p*-PGF-AA has been performed.

Figure 7 shows the TG curves of the samples of the copolymers of *p*-PGF-AA with the compositions of 30.05:69.95 mol.% (a), 45.13:54.87 mol.% (b), and 61.21:39.79 mol.% (c).

As follows from Figure 7, thermal decomposition of the copolymers proceeds in several stages. In this case, within the temperature interval of 30~140 °C, the samples do not undergo transformations, which leads to the change of their mass.

The process of degradation of the copolymer of *p*-PGF-AA with the composition of 30.05:69.95 mol.% starts at a temperature of 140 °C, and the further increase of temperature until 345 °C results in 30% mass loss of the sample. Then, the intensive decomposition of the sample is continued, ending completely at 530 °C with the formation of solid residue of about ~11%. The DTG derivative curve shows the main stages of degradation in more detail. Thus, at a temperature of 230 °C, the rate of decomposition starts to rise abruptly and reaches its maximum at 290 °C, which characterizes the second stage of degradation. On the third stage, the broad peak with a maximum rate of breaking down of about ~7%/min. is observed.

As it is seen from Figure 7b, the character of the TG curve of the copolymer of *p*-PGF-AA with the composition of 45.13:54.87 mol.% is similar to the curve of the sample with the content of 30.05:69.95 mol.%; however, it has the shift to the range of higher temperatures. On the first stage of decomposition at a temperature range of 155–250 °C, a negligible change of mass takes place, which is confirmed by a slight bend of the DTG curve. Further, within the temperature interval of 250–340 °C, there is a mass loss of about ~10% out of total mass. Starting from 340 °C, the sample begins to decompose intensively until the end of the process at 550 °C, forming a solid residual of about ~15%. From TG analysis, it follows that the copolymer with the composition of 61.21:39.79 mol.% (Figure 7c) was more thermostable. Thus, the first stage of degradation of the sample takes place within the range of ~190–285 °C. Then, active breaking down of the copolymer is observed until reaching 370 °C. On the thermogram, one can see a gradual bending of the curve at a pointed temperature. After that, there is a third stage of thermal decomposition of the sample until it finishes completely at 560 °C. The thermogravimetric data obtained during the process of thermal degradation are presented in Table 4.

The results of the kinetic investigations of thermal decomposition of the copolymers were obtained using the iso-conversional principle. This approach supposes the use of the thermogravimetric curves obtained at various temperature programs. For this purpose, the above shown TG curves at four different heating rates have been obtained and the activation energies have been calculated based on them. As an example, the graphs of linearization on four points for the copolymer of *p*-PGF-AA with the composition of 45.13:54.87 mol.% (Figure 8) are shown.

As seen from the graphs, the straight lines obtained using the differential method of Friedman have wider scattering than the ones obtained using the integral method (Figure 8).

In Figure 9, the activation energies for each separately taken conversion degree (α) within the range of 0.05–0.95 are given.

From Figure 9, it is clear that in the first stage of breaking down, the value of activation energy is around ~100 kJ/mol and it grows gradually with the increase of conversion degree. In the second stage, growth of E_a_ after reaching α = 0.6 is observed. The third stage is peculiar due to the rise of activation energy until standing at ≈250 kJ/mol, then it has meaning within the range of α = 0.4–0.8, reaching its maximum value on the final point.

The values of activation energy of thermal decomposition of the copolymers of *p*-PGF-AA of three different compositions are presented in Table 5.

Comparative analysis of obtained values of activation energies of thermal decomposition of the copolymers of *p*-PGF-AA of various molar composition (Table 5) has shown that at the initial stages the regularity is not observed and E_a_ has low meanings. At the third stage, the activation energy increases expectedly with decreasing of the amount of AA. The summative activation energy is the most for the copolymer with the higher content of unsaturated polyester. As it is obvious from Table 5, the values obtained from differential and integral methods have good correlation. Thus, the methods of Kissinger–Akakhira–Sunose and Friedman allowed us to evaluate the activation energies within the whole process before preliminary use of the models of the reactions.

## 4. Conclusions

Thus, the study of heat effect and residual reactivity of the curing process of *p*-PGF-AA using the DSC method allowed for defining the kinetic parameters of the reaction. Hence, it has been established that variation of composition of the initial polymer–monomer mixture and reaction conditions gives opportunity to control the curing process of *p*-PGF-AA and to obtain the copolymers with given properties.

On the basis of SEM data, it has been found that the samples with high content of *p*-PGF had more denser and monolithic structure with average pore size from 250 nm to 143 nm.

The methods of thermogravimetric analysis have shown the increase of thermostability of cured samples with the growth of concentration of *p*-PGF.

## Figures and Tables

**Figure 1 polymers-15-03753-f001:**
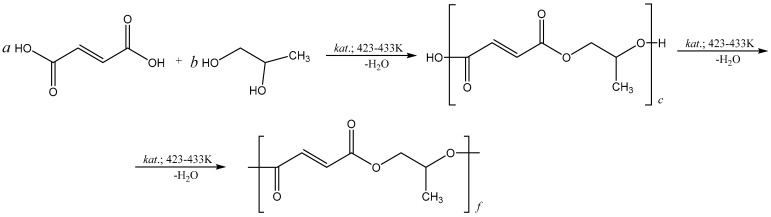
The scheme of formation of polypropylene glycol fumarate where c = a + b (the formation of acidic ester); f = a + b (the formation of polyester).

**Figure 2 polymers-15-03753-f002:**
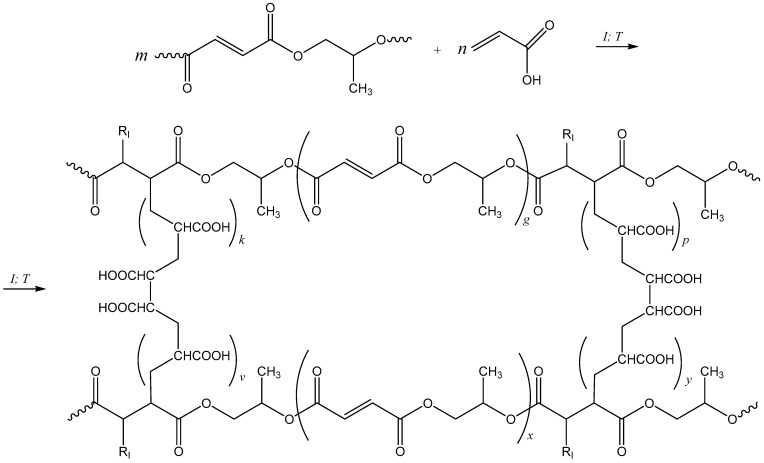
Synthesis of the copolymers of *p*-PGF with AA. R_I_ is the radical of initiator.

**Figure 3 polymers-15-03753-f003:**
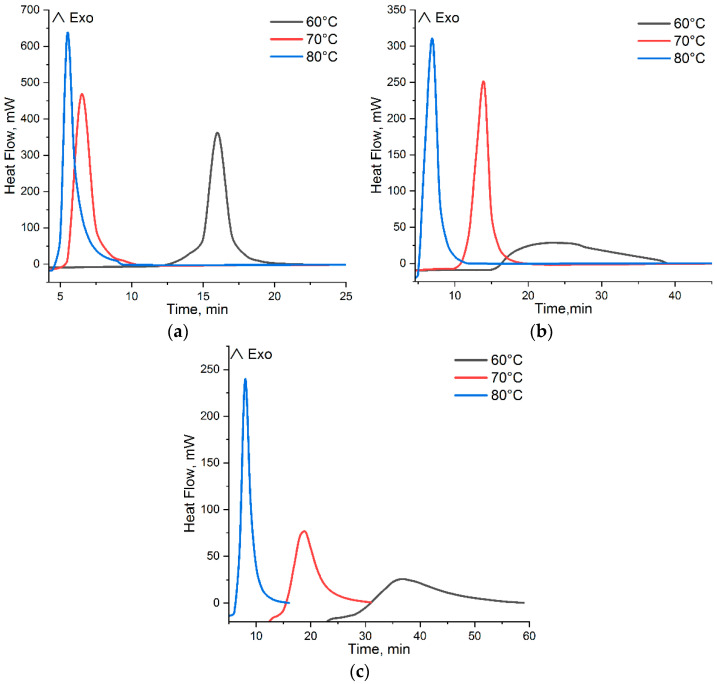
Isothermal DSC thermograms of curing *p*-PGF-AA: 30.05:69.95 mol.% (**a**); 45.13:54.87 mol.% (**b**); 61.21:39.79 mol.% (**c**).

**Figure 4 polymers-15-03753-f004:**
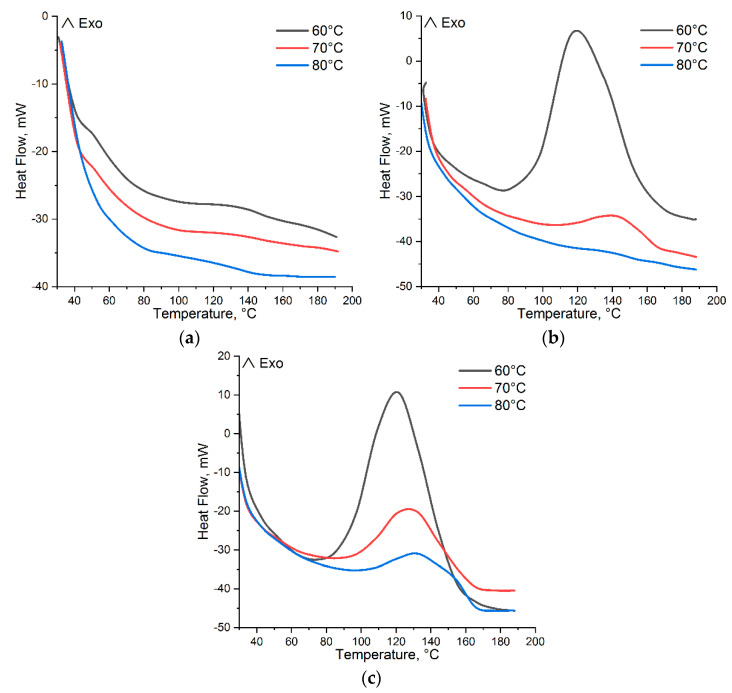
Dynamic DSC thermograms after isothermal curing of *p*-PGF-AA: 30.05:69.95 mol.% (**a**); 45.13:54.87 mol.% (**b**); 61.21:39.79 mol.% (**c**).

**Figure 5 polymers-15-03753-f005:**
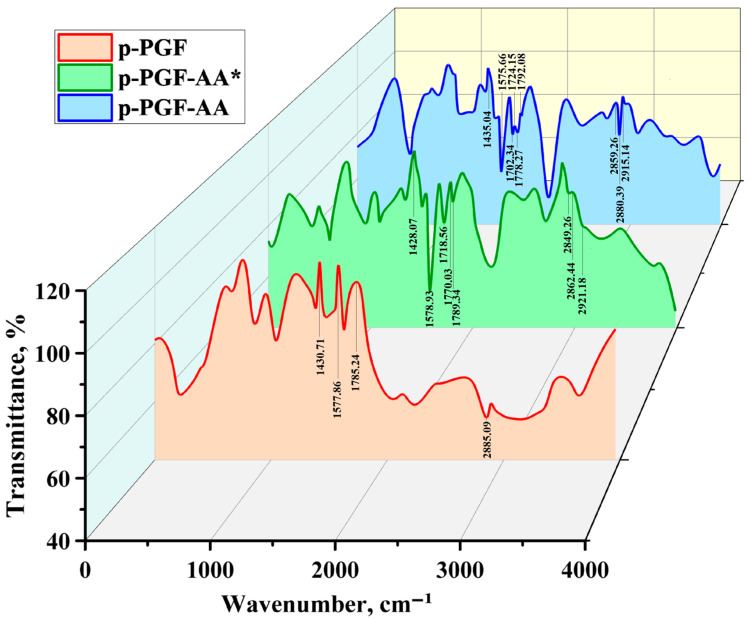
IR-spectra of *p*-PGF and the copolymer of *p*-PGF-AA of the composition of 45.13:54.87 mol.%.

**Figure 6 polymers-15-03753-f006:**
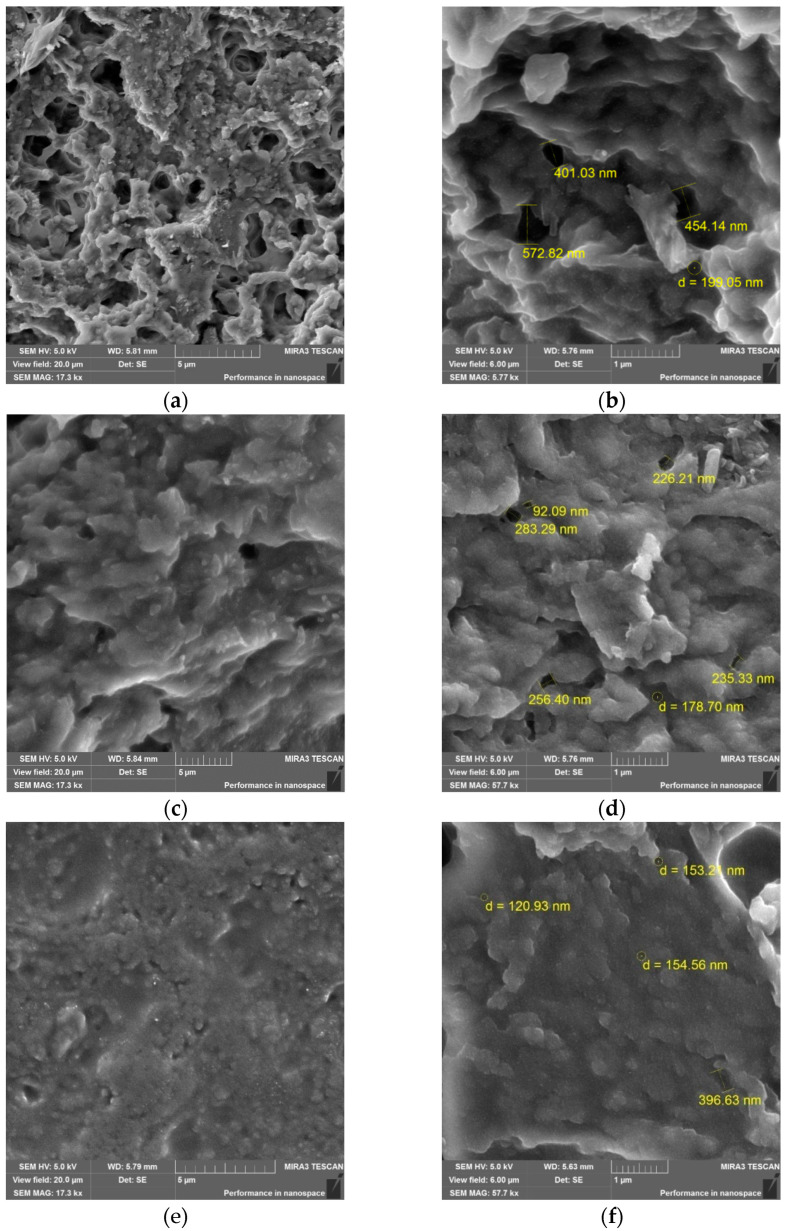
SEM images of *p*-PGF-AA of the composition of 30.05:69.95 mol.% (**a**,**b**); 45.13:54.87 mol.% (**c**,**d**); 61.21:39.79 mol.% (**e**,**f**).

**Figure 7 polymers-15-03753-f007:**
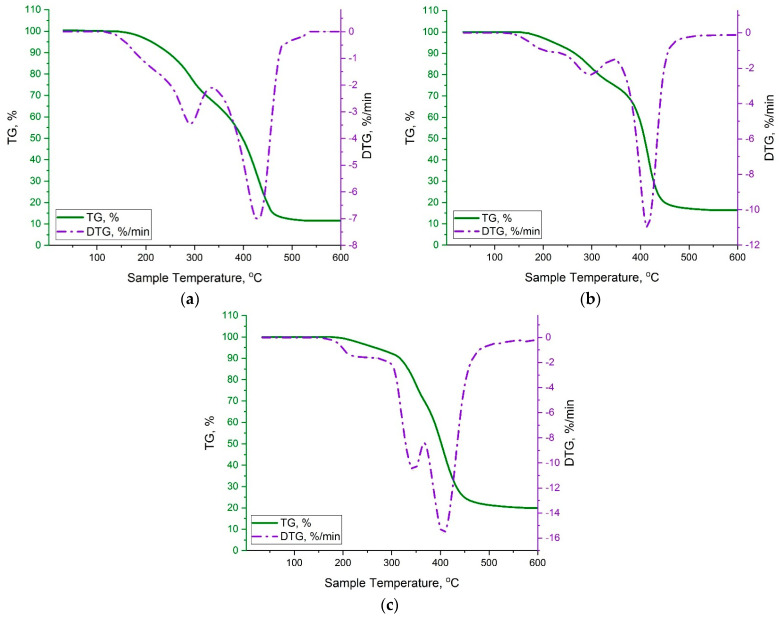
TG-curves of the copolymers of *p*-PGF-AA at a heating rate of 10 °C/min in inert medium: 30.05:69.95 mol.% (**a**); 45.13:54.87 mol.% (**b**); and 61.21:39.79 mol.% (**c**).

**Figure 8 polymers-15-03753-f008:**
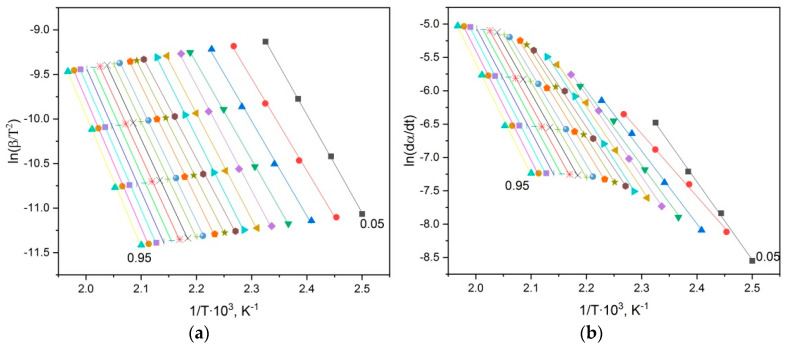
Graphical dependences of the equations of Kissinger–Akakhira–Sunose (**a,c,e**) and Friedman (**b,d,f**) for the copolymer of *p*-PGF-AA of composition 45.13:54.87 mol.%: (**a,b**) I stage, (**c,d**) II stage, (**e,f**) III stage.

**Figure 9 polymers-15-03753-f009:**
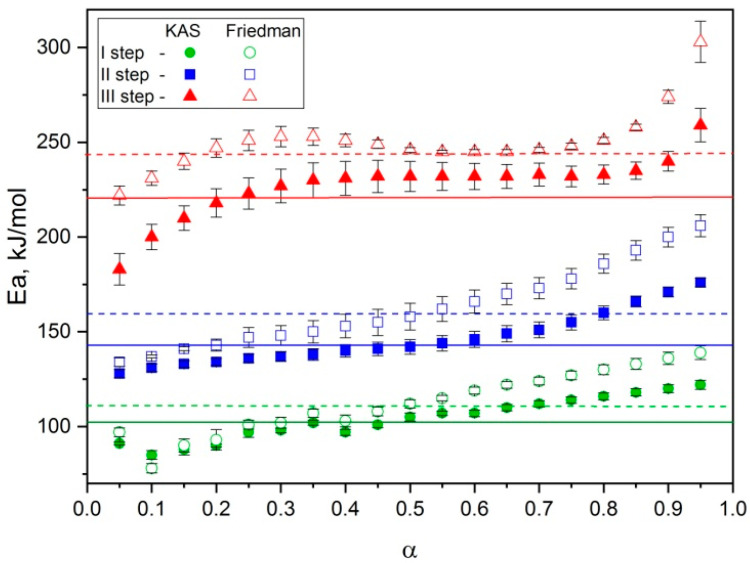
Dependence of the activation energy (E_a_) on the conversion degree (α) for the copolymer *p*-PGF-AA 45.13:54.87 mol.%.

**Table 1 polymers-15-03753-t001:** Induction time (t_i_), curing time (t_curing_), and vitrification time (t_v_) at isothermal curing of the samples of *p*-PGF-AA.

Sample, mol.%		Temperature, °C	t_i_, min	t_curing_, min	t_v_, min
*p*-PGF	AA				
30.05	69.95	60	11.30	15.10	21.30
		70	04.40	05.70	10.20
		80	04.40	05.27	09.40
45.13	54.87	60	15.10	24.40	39.80
		70	09.50	13.48	19.10
		80	04.80	06.24	10.90
61.21	39.79	60	27.40	36.28	59.00
		70	13.40	18.71	31.40
		80	05.70	08.01	16.40

**Table 2 polymers-15-03753-t002:** Reaction heats obtained from isothermal and dynamic DSC curve.

Sample, mol.%		Temperature, °C	−∆H_iso_, J/g	−∆H_r_, J/g	−∆H_tot_, J/g
*p*-PGF	AA				
30.05	69.95	60	374.318	1.687	376.005
		70	393.180	0.827	394.007
		80	410.006	0.078	410.084
45.13	54.87	60	286.880	51.115	337.995
		70	354.397	8.894	363.291
		80	377.947	0.933	378.880
61.21	39.79	60	226.339	98.605	324.944
		70	308.479	37.599	346.078
		80	330.987	22.098	353.085

**Table 3 polymers-15-03753-t003:** Kinetic parameters of the reaction of cure of *p*-PGF-AA, [BP] = 1% on a mass of initial mixture.

Sample, mol.%		Temperature, °C	α, %	k/10^3^, s^−1^	E_a_, kJ/mol
*p*-PGF	AA				
30.05	69.95	60	99.55	6.13	27.4
		70	99.79	8.01	
		80	99.98	10.73	
45.13	54.87	60	84.87	3.06	48.4
		70	97.55	5.85	
		80	99.75	8.25	
61.21	39.79	60	69.66	1.11	77.3
		70	89.14	2.79	
		80	93.74	5.41	

**Table 4 polymers-15-03753-t004:** Thermogravimetric data for *p*-PGF-AA copolymers at various heating rates.

Sample, mol.%		Temperature Range, °C			dTGmax, %/min	Residue, %
*p*-PGF	AA	I Stage	II Stage	III Stage		
30.05	69.95	140–230	230–345	345–530	7	11
45.13	54.87	155–250	250–340	340–550	11	15
61.21	39.79	190–285	285–370	370–560	16	20

**Table 5 polymers-15-03753-t005:** Activation energies of thermal decomposition of the copolymers of *p*-PGF-AA.

Sample, mol.%		Ea_ave_, kJ/mol			
*p*-PGF	AA	I Stage	II Stage	III Stage	Σ
Method of Kissinger–Akakhira–Sunose					
30.05	69.95	140 ± 5.3	117 ± 3.7	199 ± 6.1	456 ± 15.1
45.13	54.87	104 ± 1.8	146 ± 3.1	227 ± 7.2	477 ± 12.1
61.21	39.79	98 ± 1.7	140 ± 2.3	270 ± 9.5	508 ± 13.5
Method of Friedman					
30.05	69.95	140 ± 7.7	120 ± 5.6	226 ± 8.1	486 ± 21.4
45.13	54.87	112 ± 2.6	163 ± 4.9	250 ± 3.4	525 ± 10.9
61.21	39.79	112 ± 3.1	164 ± 3.4	272 ± 12.2	548 ± 18.7

## Data Availability

Not applicable.

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
