# Peer review of "Investigation of Curing Process and Thermal Behavior of Copolymers Based on Polypropylene Glycol Fumarate and Acrylic Acid Using the Methods of DSC and TGA"

_polymers, 2023, doi:10.3390/polym15183753_

Round 1

Reviewer 1 Report

This manuscript investigated the curing process of polypropylene glycol fumarate with acrylic acid using DSC and the thermal properties of these copolymers with TGA. The study filled the empty area of the curing process of unsaturated polyesters with unsaturated carbocyclic acids studied by DSC, as well as their thermal stability.

Here are my comments and suggestions:
1. In line 57, please change thermosTable to thermostable.

2. In the section on materials, please add the vendor, purity, and any other needed information for the chemicals, including reagents and solvents.

3. In Figure, please also explain c on the repeat units.

4. In GPC methods, please explain what kind of MW was reported in the manuscript: Mn or Mw. How was the MW of the sample calculated? Is it obtained by light scattering (LS), conventional calibration (CC), or universal calibration (UC)? If it is obtained by LS, please report dn/dc of the sample. If it is obtained by CC, please add column information, flow rate, temperature, and so on.

5. In line 96, please explain how the DSC cooled quickly, for example, by liquid nitrogen cooling, by water (water temperature should be reported), or by air cooling.

6. To report the sample with different mol% of p-PGF and AA, instead of listing the composition several times in the manuscript, it may be better to rename the samples because the long names are repeated too many times. For example, rename the sample of the composition of p-PGF-AA 45.13:54.87 mol.% as PGF45 or PGF45AA55. Then, the statement in line 234 can be rewritten as: “Thus, PGF45 at a temperature of 60°Ð¡ had a ~85% conversion, while the degree of conversion goes up and reaches the values up to ~97% and 99% at 70°Ð¡ and 80°Ð¡, correspondingly.”

Author Response

Response to Reviewer 1 Comments

The authors express gratitude to the reviewer for the interest to the article and for the comments. According to the comments of the reviewer the authors have made corrections to the text of the article

Point 1: In line 57, please change thermosTable to thermostable.

Response 1: This comment is taken into account and corrected in the new version of the article.

Point 2: In the section on materials, please add the vendor, purity, and any other needed information for the chemicals, including reagents and solvents.

Response 2: We are thankful for this comment. In the section “Materials” the information according to the comments of the reviewer is added.

Point 3: In Figure, please also explain c on the repeat units.

Response 3: The authors has given the scheme of the synthesis of p-PGF in Fig. 1. According to the reaction scheme, the first elementary stage of the interaction of fumaric acid with glycol is the reaction of esterification with the formation of acidic ether (c).This comment has been added to the description of Fig.1.

Point 4: In GPC methods, please explain what kind of MW was reported in the manuscript: Mn or Mw. How was the MW of the sample calculated? Is it obtained by light scattering (LS), conventional calibration (CC), or universal calibration (UC)? If it is obtained by LS, please report dn/dc of the sample. If it is obtained by CC, please add column information, flow rate, temperature, and so on.

Response 4: In the new version of the manuscript the method of GPC is suppllemented according to the comments of the reviewer.

Point 5: In line 96, please explain how the DSC cooled quickly, for example, by liquid nitrogen cooling, by water (water temperature should be reported), or by air cooling.

Response 5: The authors have added the procedure of determination of the residual heat, in particular, the temperature of water and the rate of cooling have been given.

Point 6: To report the sample with different mol% of p-PGF and AA, instead of listing the composition several times in the manuscript, it may be better to rename the samples because the long names are repeated too many times. For example, rename the sample of the composition of p-PGF-AA 45.13:54.87 mol.% as PGF45 or PGF45AA55. Then, the statement in line 234 can be rewritten as: “Thus, PGF45 at a temperature of 60°Ð¡ had a ~85% conversion, while the degree of conversion goes up and reaches the values up to ~97% and 99% at 70°Ð¡ and 80°Ð¡, correspondingly.”

Response 6: We are greateful to the reviewer. Indeed, the suggested versions of the name of the samples are suitable, nevertheless, the authors would like to keep the initial names of the samples pointing  their compositions. The change of the names in many cases leads to some difficulties in understanding of the discussion of obtained results and requires making the essential corrections in the description of the results in the text of manuscript.

This sentence has been edited to your recommendations: “Thus, PGF45 at a temperature of 60°Ð¡ had a ~85% conversion, while the degree of conversion goes up and reaches the values up to ~97% and 99% at 70°Ð¡ and 80°Ð¡, correspondingly.”

Reviewer 2 Report

Interesting paper. This paper studies the curing process of polypro- 60 pylene glycol fumarate with acrylic acid using DSC method and thermal stability using TGA. IR and SEM are also applied.

Some comments:

1. Font of some equations could be larger. For example, font of eqn 1 &3 are too small.

2. The transmittance values in Figure 5 are too small. 

3. Figure 7 (a) is not complete. Please adjust the figure size.

4. For figure 4, in order to have more convincing results of residual cure, I will suggest to use heat-cool-heat procedure.

5. There is a large blank space in page 13. Please adjust the content.

6. Figure 6 is not clear for me. Have you polished the samples? Is it polished surface or unpolished surface?

Author Response

Response to Reviewer 2 Comments

The authors express gratitude to the reviewer for the interest to this article and for the comments. The authors have taken into consideration all the comments of the reviewer in a new version of the article.

Point 1: Font of some equations could be larger. For example, font of eqn 1 &3 are too small.

Response 1: The authors have changed the font of the equations.

Point 2: The transmittance values in Figure 5 are too small. 

Response 2: This comment is taken into consideration and corrected in the new version of the article.

Point 3: Figure 7 (a) is not complete. Please adjust the figure size.

Response 3: In the new version of the article the Fig. 7 is adjusted.

Point 4: For figure 4, in order to have more convincing results of residual cure, I will suggest to use heat-cool-heat procedure.

Response 4: The authors completely agree with the reviewer and have introduced the comment to the text to Figure 4: “Thus, determination of the residual heat was carried out after isothermal cure of p-PGF-AA by cooling the system followed by its heating at dynamic regimens (the “heat-cool-heat” procedure).”. Besides, in the section “Methods” the authors have added the procedure of determination of the residual heat.

Point 5: There is a large blank space in page 13. Please adjust the content.

Response 5: This comment is taken into account and corrected in the new version of the article.

Point 6: Figure 6 is not clear for me. Have you polished the samples? Is it polished surface or unpolished surface?

Response 6: We are thankful for the comment. SEM analysis has shown the growth of the density of polymer network with the increase of the amount of p-PGF. SEM analysis has been carried out on bucked/powdered copolymers of p-PGF-AA.
